# A DYNAMIC GAME APPROACH TO TRAINING ROBUST DEEP POLICIES.

## ABSTRACT

We present a method for evaluating the sensitivity of deep reinforcement learning (RL) policies. We also formulate a zero-sum dynamic game for designing robust deep reinforcement learning policies. Our approach mitigates the brittleness of policies when agents are trained in a simulated environment and are later exposed to the real world where it is hazardous to employ RL policies. This framework for training deep RL policies involve a zero-sum dynamic game against an adversarial agent, where the goal is to drive the system dynamics to a saddle region. Using a variant of the guided policy search algorithm, our agent learns to adopt robust policies that require less samples for learning the dynamics and performs better than the GPS algorithm. Without loss of generality, we demonstrate that deep RL policies trained in this fashion will be maximally robust to a "worst" possible adversarial disturbances.

## 1  INTRODUCTION

Deep reinforcement learning (RL) for complex agent behavior in realistic environments usually combines function approximation techniques with learning-based control. A good RL controller should guarantee fulfillment of performance specifications under external disturbances, or modeling errors. Quite often in practice, however, this is not the case – with deep RL policies not often generalizing well to real-world scenarios. This can be attributed to the inherent differences between the training and testing environments. Recently, there have been efforts at integrating function approximation techniques with learning-based control, in an end-to-end fashion, in order to have systems that optimize objectives while guaranteeing generalization to environmental uncertainties. Examples include trajectory-based optimization for known dynamics ([17, 26]), or trajectory optimization for unknown dynamics such as guided policy search algorithms [1, 14, 16].

While these methods produce performance efficiency for agent tasks in the real world, there are sensitivity questions of such policies that need to be addressed such as, how to guarantee maximally robust deep RL policies in the presence of external disturbances, or modeling errors. A typical approach employed in minimizing sample inefficiency is to engineer an agent's policy in a simulated environment, and later transfer such policies to physical environments. However, questions of robustness persist in such scenarios as the agent often has to cope with modeling errors and new sensory inputs from a different environment. For continuous control tasks, learned policies may become brittle in the presence of external perturbations, or a slight change in the system dynamics may significantly affect the performance of the learned controller [21] – defeating the purpose of having a robust policy that is learned through environmental interaction .

The contribution of this paper is two-fold:

- first, we provide a framework that demonstrates the brittleness of a state-of-the-art deep RL policy; specifically, given a trained RL policy, we pose an adversarial agent against the fixed trained policy; the goal is to perturb the parameter space of the learned policy. We demonstrate that the most sophisticated deep policies fail in the presence of adversarial perturbations.

- second, we formulate an iterative dynamic zero-sum, two player game, where each agent executes an opposite reaction to its pair: a concave-convex problem follows explicitly, and

our goal is to achieve a saddle point equilibrium, where the state is everywhere defined but possibly infinite-valued).

Noting that lack of generalization of learned reward functions to the real-world can be thought of as external disturbance that perturb the system dynamics, we formulate the learning of robust control policies as a zero-sum two player Markov game – an iterative dynamic game (iDG) – that pits an adversarial agent against a protagonist agent.

The controller aims to minimize a given cost while the second agent, an adversary aims to maximize the given cost in the presence of an additive disturbance. We run the algorithm in finite episodic settings and show a dynamic game approach aimed at generating policies that are maximally robust.

The content of this paper is thus organized: we review relevant literature to our contribution in Sec. 2; we then provide an $H_\infty$ background in Sec. 3. This $H_\infty$ technical introduction will be used in formulating the design of perturbation signals in Sec. 4. Without loss of generality, we provide a formal treatment of the iDG algorithm within the guided policy search framework in Sec. 5. Experimental evaluation on multiple robots is provided in Sec. 6 followed by conclusions in Sec. 7.

## 2 RELATED WORK

Robustness studies in classical control have witnessed the formalization of algorithms and computation necessary to carry out stable feedback control and dynamic game tasks (e.g. [2, 15, 19]). There now exist closed-form and iterative-based algorithms to quantify the sensitivity of a control system and design robust feedback controllers. These methods are well-studied in classical $H_\infty$ control theory. While questions of robustness of policies have existed for long in connectionist RL settings[25], only recently have researchers started addressing the question of incorporating robustness guarantees into deep RL controllers.

Heess et. al [9] posit that rich, robust performance will emerge if an agent is simulated in a sufficiently rich and diverse environment. [9] proposed a learning framework for agents in locomotion tasks which involved choosing simple reward functions, but exposing the agent to various levels of difficult environments as a way of achieving *ostensibly sophisticated* performance objectives. Incorporating various levels of difficulty in obstacles, height and terrain smoothness to an agent's environment for every episodic task, they achieved robust behaviors for difficult locomotion tasks after many episodes ($\approx 10^6$) of training. However, this strategy defeats one of the primary objectives of RL namely, to make an agent discover good policies with finite data based on little interaction with the environment. An ideal robust RL controller must come from data-efficient samples or imitations. Furthermore, this approach takes a qualitative measure at building robust signals into the reward function via means such as locomotion hurdles with variations in height, slopes, and slalom walls. We reckon that building such physical barriers for an agent is expensive in the real-world and learning such emergent locomotion behaviors takes a long training time.

Pinto et. al. [20], posed the learning of robust RL rewards in a zero-sum, two-player markov decision process (MDP) defined by the standard RL tuple $\{\mathcal{S}, \mathcal{A}_1, \mathcal{A}_2, \mathcal{P}, \mathcal{R}, \gamma, s_0\}$, where $\mathcal{A}_1$ and $\mathcal{A}_2$ denote the continuous action spaces of the two players. Both players share a joint transition probability $\mathcal{P}$ and reward $\mathcal{R}$. Pinto's approach assumed a knowledge of the underlying dynamics so that an adversarial policy, $\pi_\theta^{\mathrm{adv}}(\mathbf{u}_t|\mathbf{x}_t)$, can exploit the weakness in a protagonist agent's policy, $\pi_\theta^{\mathrm{prot}}(\mathbf{u}_t|\mathbf{x}_t)$. This relied on a minimax alternating optimization process: optimizing for one set of actions while holding the other fixed, to the end of ensuring robustness of the learned reward function. While it introduced $H_\infty$ control as a robustness measure for classical RL problems, it falls short of adapting $H_\infty$ for complex agent tasks and policy optimizations. Moreover, there are no theoretical analyses of saddle-/pareto-point or Nash equilibrium guarantees and the global optimum that assures maximal robustness at $\pi_\theta^{\mathrm{prot}}(\mathbf{u}_t|\mathbf{x}_t) = \pi_\theta^{\mathrm{adv}}(\mathbf{u}_t|\mathbf{x}_t)$ is left unaddressed.

Perhaps the closest formulation to this work is [10]'s neural fictitious self-play for large games with imperfect state information whereby players select best responses to their opponents' average strategies. [10] showed that with deep reinforcement learning, self-play in imperfect-information environments approached a Nash equilibrium where other reinforcement learning methods diverged.

From a methodical perspective, we formulate the robustness of RL controllers within an $H_\infty$ framework (see [19]) for deep robot motor tasks. Similar to a matrix game with two agents, we let both agents play a zero-sum, two person game where each agent's action strategy or *security level* never falls below that of the other. The ordering according to which the players act so that each player acts optimally in a "min max" fashion does not affect the convergence to saddle point in our formulation. We consider the case where the security levels of both players coincide so that the strategy pair of both agents constitute a *saddle-point pure strategy* [3, p. 19].

## 3 BACKGROUND AND PRELIMINARIES

Reinforcement learning in robot control tasks consists of selecting control commands $\mathbf{u}$ from the space of a control policy $\pi$ (often parameterized by $\theta$) that act on a high-dimensional state $\mathbf{x}$. The $\mathbf{x}$ typically composed of internal (e.g. joint angles and velocities) and external (e.g. object pose, positional information in the world) components. For a stochastic policy $\pi(\mathbf{u}_t|\mathbf{x}_t)$ the commands influence the state of the robot based on the transition distribution $\pi_\theta(\mathbf{u}_t|\mathbf{x}_t, t)$. The state and action pairs constitute a trajectory distribution $\tau = (\mathbf{x}_1, \mathbf{u}_1, \mathbf{x}_2, \mathbf{u}_2, \cdots, \mathbf{x}_T, \mathbf{u}_T)$.

The performance of the robot on an episodic motor task is evaluated by an accumulated reward function $\mathcal{R}(\tau)$ defined as

$$\mathcal{R}(\tau) = \sum_{t=0}^{T-1} r_t(\mathbf{x}_t, \mathbf{u}_t) + r_{t_f}(\mathbf{x}_{t_f})$$

for an instantaneous reward function, $r_t$, and a final reward, $r_{t_f}$. Many tasks in robot learning domains can be formulated as above, whereby we choose a locally optimal policy $\pi_\theta^\star$ that optimizes the expectation of the accumulated reward

$$\ell_{\pi_\theta} = \mathbb{E}(\mathcal{R}(\tau)|\pi_\theta) = \int \mathcal{R}(\tau) p_{\pi_\theta}(\tau) d\tau,$$

where $p_{\pi_\theta}(\tau)$ denotes distribution over trajectories $\tau$ and is defined as

$$p_{\pi_\theta}(\tau) = p(x_1) \prod_{t=0}^{T-1} \pi_\theta(\mathbf{u}_t|\mathbf{x}_t) p(\mathbf{x}_{t+1}|\mathbf{x}_t, \mathbf{u}_t).$$

$p(\mathbf{x}_{t+1}|\mathbf{x}_t, \mathbf{u}_t)$ above represents the robot's dynamics and its environment.

Given the inadequacy of value function approximation methods in managing high-dimensional continuous state and action spaces as well as the difficulty of carrying out arbitrary exploration given hardware constraints [7], we resolve to use policy search (PS) methods as they operate in the parameter space of parameterized policies. However, direct PS are often specialized algorithms that produce optimal policies for a particular task (often using policy gradient methods), and they come with the negative effects of not generalizing well to flexible trajectory optimizations and large representations e.g. using neural network policies.

Guided policy search (GPS) algorithms [1, 14, 16] are able to guide the search for parameterized policies from poor local minima using an alternating block coordinate ascent of the optimization problem, made up of a *C-Step* and an *S-Step*. In the **C-step**, a well-posed cost function is minimized with respect to the trajectory samples, generating guiding distributions $p_i(\mathbf{u}_t|\mathbf{x}_t)$; and in the **S-step**, the locally learned time-varying control laws, $p_i(\mathbf{u}_t|\mathbf{x}_t)$, are parameterized by a nonlinear, neural network policy using supervised learning. The S-step fits policies of the form $\pi_\theta(\mathbf{u}_t|\mathbf{x}_t) = \mathcal{N}(\mu^\pi(\mathbf{x}_t), \Sigma^\pi(\mathbf{x}_t))$ to the local controllers $p_i(\mathbf{u}_t|\mathbf{x}_t)$, where $\mu^\pi(\mathbf{x}_t)$, and $\Sigma^\pi(\mathbf{x}_t)$ are functions that are estimated.

In order to ensure the learned policy for a dynamical system is robust to external uncertainties, modeling and transfer learning errors, we propose an iterative dynamic game consisting of an agent within an environment, and an adversarial agent, interacting with the original agent in the closed-loop environment $\mathcal{E}$, over a finite horizon, $T$ (it could also be extended to the infinite horizon case). The adversary could represent a spoofing agent in the world or modeling errors between the plant and dynamics. The states evolve according to the following stochastic dynamics $p(\mathbf{x}_{t+1}|\mathbf{x}_t, \mathbf{u}_t, \mathbf{v}_t), \forall t = 0, ..., T$ where $\mathbf{x}_t \in \mathcal{X}_t$ is a markovian system state, $\mathbf{u}_t \in \mathcal{U}_t$ is the action taken by the agent (henceforth called the protagonist), $\mathbf{v}_t \in \mathcal{V}_t$ is the action taken by an

adversarial agent. The subscripts denote time steps $t \in [1, T]$, allowing for a simpler structuring of the individual policies per time step [7]. The problems we consider are control tasks with complex dynamics, having continuous state and action spaces, and with trajectories defined as $\bar{\tau} = \{\mathbf{x}_1, \mathbf{u}_1, \mathbf{v}_1, \mathbf{x}_2, \mathbf{u}_2, \mathbf{v}_2, \cdots, \mathbf{x}_{t_f}, \mathbf{u}_{t_f}, \mathbf{v}_{t_f}\}$. At time $t$, the system's controller visits states with high rewards while the adversarial agent wants to visit states with low rewards. The solution to this zero-sum game follows with an equilibrium at the origin – a *saddle-point* solution ensues.

The policies that govern the behavior of the agents are defined as $\boldsymbol{\pi_\theta}(\mathbf{u}_t|\mathbf{x}_t)$ and $\pi_\theta(\mathbf{v}_k|\mathbf{x}_k)$ respectively, and the learned local linear time-varying Gaussian controllers are defined as $p(\mathbf{u}_k|\mathbf{x}_k)$, $p(\mathbf{v}_k|\mathbf{x}_k)$. In the next subsection, we show that learned motor policies, $p(\mathbf{u}_k|\mathbf{x}_k)$, are sensitive to minute additive disturbances; we later on propose how to build robustness to such trained neural network policies. This is important in learning tasks where the robustness margins of a trained controller need to be known in advance before being introduced to a new execution environment.

Our goal is to select a suitable policy parameterization, so as to assure robustness and stability guarantees [4]. In this paper, we specifically use convex variant of the mirror descent version of GPS [16].

## 4 CASE FOR ROBUSTNESS IN PS ALGORITHMS

In this section, we show why guided policy search algorithms are non-robust to even the simplest form of perturbations – additive disturbance. Before we proceed, we note that policy search algorithms are popular among the RL tools available because they have a "modest" level of robustness built into them e.g.

- by requiring the learning controller to start the policy parameterization from multiple initial states,
- adding a Kullback-Leibler (KL) constraint term to the reward function e.g. [1, 14],
- solving a motor task in multiple ways where more than one solution exist [7],
- or by introducing additive noise into the system as white noise e.g. differential dynamic programming (DDP) methods [11] or their iLQG variants[23].

A fundamental drawback of these robustness mechanisms, however, is that the learned policy can only tolerate disturbance for slightly changing conditions; parametric uncertainty is not suitably modeled as white noise, and treating the error as an extra input might be relative to the size of the inputs (drawn from the environment) – necessitating the need for a formal treatment of robustness in PS algorithms.

A methodical way of solving the robustness problem in deep RL would be to consider techniques formalized in $H_\infty$ control theory, where controller sensitivity and robustness are solved within the framework of a differential game. We conjecture that the lack of robustness of a RL trained policy arises from the difference between a plant model and the real system, or the difference in learning environments. If we can measure the sensitivity of a system, $\boldsymbol{\gamma}$, then we can aim for policy robustness by ensuring that $\gamma$ is sufficiently small to reject disturbance arising

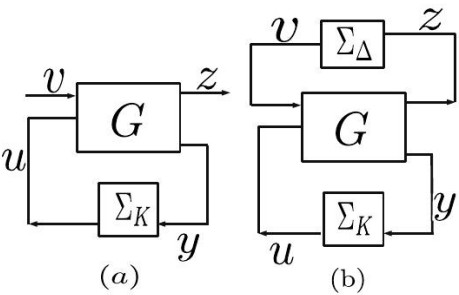

from the training environment or modeling errors if the gain of mapping from the error space to the disturbance is less than $\gamma^{-1}$[27]. The figure to the right depicts the standard $H_\infty$ control problem. Suppose $G$ in the left inset is a plant for which we can find an internally stabilizing controller, $\Sigma_K$, that ensures stable transfer of input $u$ to measurement $y$, the $H_\infty$ control objective is to find the "worst" possible disturbance, $v$, which produces an undesired output, $z$; we want to minimize the effect of $z$. In the right inset in the figure, we treat unmodeled dynamics, transfer errors and other uncertainty as an additional feedback to which we would like to adapt with respect to the worst possible disturbance in a prescribed range. $\Sigma_\Delta$ in the right inset represents these uncertainties; our goal is to find the closed-loop optimal policy for which the plant, $G$, will satisfy performance requirements and maintain robustness for a large range of systems $\Sigma_\Delta$. We focus on conditions under

---

**Algorithm 1** Guided policy search: convex linear variant

---

1: **for** iteration $k \in \{1, \ldots, K\}$ **do**
2:     **C-step**: $p_i \leftarrow \arg\min_{p_i} \mathbb{E}_{p_i(\boldsymbol{\tau})} \left[ \sum_{t=1}^{T} r(\mathbf{x}_t, \mathbf{u}_t) \right]$ such that $D_{KL}(p_i(\boldsymbol{\tau}) \| \boldsymbol{\pi_\theta}(\boldsymbol{\tau})) \leq \epsilon$
3:     **S-step**: $\boldsymbol{\pi_\theta} \leftarrow \arg\min_\theta \sum_i D_{KL}(p_i(\boldsymbol{\tau}) \| \boldsymbol{\pi_\theta}(\boldsymbol{\tau}))$ (from supervised learning)
4: **end for**

---

which we can make the $H_\infty$ norm of the system less than a given prior, $\boldsymbol{\gamma}$. Specifically, we want to design a controller $\Sigma_K$ that minimizes the $H_\infty$ norm of the closed-loop transfer function $T_{zv}$ from disturbance $v$ to output $z$ defined as $\|T_{zv}\|_\infty = \sup_v \frac{\|z\|_2}{\|v\|_2}$.

From the small-gain theorem, the system in the right figure above will be stable for any stable mapping $\Delta : z \to v$ for $\|\Delta\|_\infty < \gamma^{-1}$ [19]. In a differential game setting, we can consider a min-max solution to the $H_\infty$ problem for the plant G with dynamics given by $\dot{x} = f(x, u, w)$ so that we solve an $H_\infty$ problem that satisfies the constraint $\|T_{zv}\|_\infty = \sup_v \frac{\|z\|_2^2}{\|v\|_2^2} \leq \gamma^2$, or find a control input $u$ that satisfies the constraint $V = \int_{t=0}^{T} \left( z_t^T z_t - \boldsymbol{\gamma}^2 v_t^T v_t \right) dt \leq 0$ for all possible disturbances $v$ for which $x_0 = 0$.

We consider a differential game for which the best control $u$ that minimizes $V$, and the worst disturbance $v$ that maximizes $V$ are derived from

$$V^\star = \min_u \max_v \int_{t=0}^{T} \left[ z_t^T z_t - \boldsymbol{\gamma}^2 v_t^T v_t dt \right]. \tag{1}$$

The optimal value function is determined from the Hamilton-Jacobi-Isaacs (HJI) equation,

$$\min_u \max_v \left[ z_t^T z_t - \boldsymbol{\gamma}^2 v_t^T v_t + \frac{\partial V^\star}{\partial x} f(x, u, v) \right] = 0$$

from which the optimal $u$ and $v$ can be computed. $\epsilon$ is adjusted based on the formulation in [16]. The dynamics $p_i(\mathbf{x}_{k+1} | \mathbf{x}_k, \mathbf{u}_k, \mathbf{v}_k) = \mathcal{N}(f_{\mathbf{x}k}\mathbf{x}_k + f_{\mathbf{u}k}\mathbf{u}_k + f_{\mathbf{v}_k}\mathbf{v}_k, F_k)$ are fitted to samples $\{\mathbf{x}_{k+1}^i, \mathbf{x}_k^i, \mathbf{u}_k^i, \mathbf{v}_k^i\}$ using a mixture of Gaussian models to the generated samples $\{\mathbf{x}_{k+1}^i, \mathbf{x}_k^i, \mathbf{u}_k^i, \mathbf{v}_k^i\}$ at iteration $i$ for all time $k$. Specifically, we incorporate a normal-inverse-Wishart prior on the Gaussian model as described in [14, §A.3].

## 4.1 Sensitivity of a learned RL Policy

This section offers guidance on testing the sensitivity of a deep neural network policy for an agent. We consider additive disturbance to a deep RL policy. Our goal is to study the degradation of performance of a trained neural network policy in the presence of the "worst" possible disturbance in the parameter space of the policy; if this disturbance cannot alter the performance of the trained policy, we have some value for the policy parameters in the prescribed range that the decision strategy is acceptable. We follow the model described above, where $\Sigma_\Delta$ denotes the uncertainty injected by the adversary. We arrive at the nominal system from $\mathbf{u}$ to $\mathbf{y}$ when the transfer matrix of $\Sigma_\Delta$ is zero. We call $\Sigma_\Delta$ the adversary whose control, $\mathbf{v}$'s effect on the output $\mathbf{z}$ is to be minimized. We quantify the effect of $\mathbf{v}$ on $\mathbf{z}$ in closed loop using a suitable cost function as a min-max criteria. This can be seen as an $H_\infty$ norm on the system. Suppose the local actions, $p(\mathbf{u}_k | \mathbf{x}_k)$, of the controller belong to the policy space $\pi = [\boldsymbol{\pi}_0, ..., \boldsymbol{\pi}_T]$ that maximize the expected sum of rewards

$$\max_{p, \boldsymbol{\pi_\theta}} \mathbb{E}\left[\ell(\boldsymbol{\tau})\right] \text{ s.t. } p(\mathbf{u}_k | \mathbf{x}_k) = \boldsymbol{\pi_\theta}(\mathbf{u}_t | \mathbf{x}_t) \, \forall \, (\mathbf{x}_t, \mathbf{u}_t, t), \tag{2}$$

Therefore, the augmented reward for the closed-loop protagonist-adversary system becomes

$$\min_{p_{\mathbf{u}}, \boldsymbol{\pi}_{\theta_{\mathbf{u}}}} \max_{p_{\mathbf{v}}, \boldsymbol{\pi}_{\theta_{\mathbf{v}}}} \mathbb{E}\left[\ell(\bar{\boldsymbol{\tau}})\right] \text{ s.t. } p(\mathbf{u}_k | \mathbf{x}_k) = \boldsymbol{\pi_\theta}(\mathbf{u}_t | \mathbf{x}_t) \, \forall \, (\mathbf{x}_t, \mathbf{u}_t, \mathbf{v}_t, t), \tag{3}$$

where $\ell(\boldsymbol{\tau}) = r(\mathbf{x}_t, \mathbf{u}_t, t) + r(\mathbf{x}_{t_f}, \mathbf{u}_{t_f}, t_f)$, and $\ell(\bar{\boldsymbol{\tau}}) = r(\mathbf{x}_t, \mathbf{u}_t, \mathbf{v}_t, t) + r(\mathbf{x}_{t_f}, \mathbf{u}_{t_f}, \mathbf{v}_{t_f}, t_f)$. Essentially, $\ell(\bar{\boldsymbol{\tau}}) = \ell(\bar{\boldsymbol{\tau}}) - \gamma^2 \alpha(\mathbf{v}_t)$ where $\alpha(\cdot)$ can be chosen as a function of the adversarial disturbance $\mathbf{v}_t$[1]. We chose $\alpha$ as the $L_2$ norm of the disturbance $\mathbf{v}_t$ in our implementation. $\gamma$ is a sensitivity

---

[1]This formulation assumes that $\mathcal{V}_t$ is a vector space, though one can define nonnegative adversary input penalty functions in other settings, e.g. when $\mathcal{V}_t$ is a finite set.

parameter that adjusts the *strength* of the adversary by increasing the penalty incurred by its actions. In (2), we carry out the optimization procedure by first learning the optimal policy for the controller; we then fix this optmal policy and carry out the minimization of the augmented reward function with the adversary in closed-loop as in (3).

As $\gamma \to \infty$ in (3), the optimal closed-loop policy is for the agent to do nothing, since any action will incur a large penalty; as $\gamma$ decreases, however, the adversary's actions have a greater effect on the state of the closed-loop system. The (inverse of the) lowest value of $\gamma$ for which the adversary's policy causes unacceptable performance provides a measure of robustness of the control policy $\pi_{\boldsymbol{\theta}}(\mathbf{u}_t|\mathbf{x}_t)$. For various values of $\gamma$, the state-of-the-art robot learning policies are non-robust to small perturbations as we show in Sec. 6.

## 4.2 ROBUST ZERO-SUM, TWO-PERSON GAMES

To learn robust policies, we run an alternating optimization algorithm that maximizes the cost function with respect to the adversarial controller (modeled with the worst possible disturbance) and minimizes the cost function with respect to the protagonist's policy. We consider a two-player, zero-sum Markov game framework for simultaneously learning policies for the protagonist and the adversary. We seek to learn saddle-point equilibrium strategies for the zero-sum game:

$$\min_{p_{\mathbf{u}_t} \in \pi(\mathbf{x}_t)} \max_{p_{\mathbf{v}_t} \in \bar{\pi}(\mathbf{x}_t)} \mathbf{E} \sum_{t=0}^{T-1} \ell_t(\mathbf{x}_t, \mathbf{u}_t, \mathbf{v}_t), \tag{4}$$

where we have overloaded notation such that $\pi(\mathbf{x}_t) = \boldsymbol{\pi_{\theta}}(\mathbf{u}_t|\mathbf{x}_t)$ and $\bar{\pi}(\mathbf{x}_t) = \boldsymbol{\pi_{\theta}}(\mathbf{v}_t|\mathbf{x}_t)$; $\ell(\mathbf{x}_t, \mathbf{u}_t, \mathbf{v}_t) = r(\mathbf{x}_t, \mathbf{u}_t) + \gamma \alpha(\mathbf{v}_t)$ is the stage cost. $\bar{\pi}(\mathbf{x}_t)$ denotes that the adversarial actions are drawn from outside of the action space of the protagonist's policy. Fixing a value of $\gamma$ is equivalent to an assumption on the capability of the adversary or the magnitude of a worst possible disturbance. To validate this proposal, we develop locally robust controllers for a trajectory optimization problem from multiple initial states using (4) as a guiding cost; a neural network function approximator is then used to parameterize these local controllers using supervised learning. We discuss this procedure in the next section.

## 4.3 ROBUST GPS

GPS adds off-policy guiding samples to a sample set: this guides the policy toward spaces of high rewards. If $p(\boldsymbol{\tau})$ is the trajectory distribution induced by the locally linear Gaussian controller $p(\mathbf{u}_t|\mathbf{x}_t)$ and $\bar{p}(\mathbf{u}_t|\mathbf{x}_t)$ denotes the previous local controller, GPS algorithms reduce the effect of visiting regions of low entropy by minimizing the KL divergence of the current local policy from the previous one as follows,

$$D_{KL}\left(p(\boldsymbol{\tau}) \| \bar{p}(\boldsymbol{\tau})\right) = \mathbb{E}[-r(\tau)] - \eta \mathcal{H}(\bar{p}) \tag{5}$$

where $\mathcal{H}$ is the entropy term that favors broad distributions, $\eta$ is a Lagrange multiplier and the first term forces the actions $p$ to be high in regions of high reward. The trajectory is optimized using optimal control principles under linear quadratic Gaussian assumptions [23]. GPS minimizes the expected cost, $\mathbb{E}_{\pi_{\boldsymbol{\theta}}(\mathbf{x}_t, \mathbf{u}_t)} r(\mathbf{x}_t, \mathbf{u}_t)$ over the joint distribution of state and action pairs given by the marginals $\pi_{\boldsymbol{\theta}}(\boldsymbol{\tau}) = p(\mathbf{x}_1) \prod_{t=1}^{T} p(\mathbf{x}_{t+1}|\mathbf{x}_t, \mathbf{u}_t)$. GPS algorithms optimize the cost $J(\boldsymbol{\theta})$ via a split process of trajectory optimization of local control laws and a standard supervised learning to generalize to high-dimensional policy space settings. A generic GPS algorithm is shown in Algorithm 1. During the C-step, multiple local control laws, $p_i(\mathbf{u}_t|\mathbf{x}_t)$, are generated for different initial states $\mathbf{x}_1^i \sim p(\mathbf{x}_1)$. The supervised learning stage (S-step) regresses the global policy $\pi_{\boldsymbol{\theta}}(\mathbf{u}_t|\mathbf{x}_t)$ to all the local actions computed in the C-step. For unknown dynamics, one can fit $p(\mathbf{x}_{t+1}|\mathbf{x}_t, \mathbf{u}_t)$ to sampled trajectories from the trajectory distribution under $\bar{p}(\boldsymbol{\tau})$. To avoid divergence in dynamics, the difference between the current and previous trajectories are constrained by the KL divergence as in step 2 in algorithm 1.

The KL divergence $\bar{p}$ from $p$ in (5) will not optimize for a robust policy in the presence of modeling errors, changes in environment settings or disturbance as we show in the sensitivity section in subsection 4.1. To make the computed neural network policy robust to these uncertainties, we propose a zero-sum, two-person dynamic game scenario in the next section.

---

**Algorithm 2** Robust guided policy search: unknown nonlinear dynamics

---
1: **for** iteration $k \in \{1, \ldots, K\}$ **do**
2:     Generate samples $\mathcal{D}_i = \{\boldsymbol{\tau}_{i,j}\}$ by running $p_i(\mathbf{u}_k|\mathbf{x}_k)$ and $p_i(\mathbf{v}_k|\mathbf{x}_k)$ or $\pi_{\boldsymbol{\theta}i}(\mathbf{u}|\mathbf{x}_k)$ and $\pi_{\boldsymbol{\theta}i}(\mathbf{v}|\mathbf{x}_k)$
3:     Fit linear-Gaussian dynamics $p_i(\mathbf{x}_{k+1}|\mathbf{x}_k, \mathbf{u}_k, \mathbf{v}_k)$ using samples in $\mathcal{D}_i$
4:     Fit linearized protagonist policy $\pi_{\boldsymbol{\theta}i}(\mathbf{u}_k|\mathbf{x}_k)$ using samples in $\mathcal{D}_i$
5:     Regress global policies $\bar{\pi}_{\boldsymbol{\theta}i}(\mathbf{u}_k|\mathbf{x}_k)$, $\bar{\pi}_{\boldsymbol{\theta}i}(\mathbf{v}_k|\mathbf{x}_k)$ with samples in $\mathcal{D}_i$
6:     **C-step**: $p_i \leftarrow \arg\min_{p_{\mathbf{u}_i}} \max_{p_{\mathbf{v}_i}} \left[ \mathbb{E}_p(\bar{\boldsymbol{\tau}}) \sum_{k=1}^{K} l(\mathbf{x}_k, \mathbf{u}_k, \mathbf{v}_k) \right]$ s.t. $D_{KL}(p_{\mathbf{u}_i}(\bar{\boldsymbol{\tau}})\|\bar{\pi}_{\boldsymbol{\theta}_{\mathbf{u}_i}}(\bar{\boldsymbol{\tau}})) \leq \epsilon$
7:     **S-step**: $\pi_{\boldsymbol{\theta}} \leftarrow \arg\min_{\boldsymbol{\theta}} \max_{\boldsymbol{\theta}} \sum_{k,i,j} D_{KL}(\pi_{\boldsymbol{\theta}}(\mathbf{u}_k|\mathbf{x}_{k,i,j})\|p_{\mathbf{u}_i}(\mathbf{u}_k|\mathbf{x}_{k,i,j}))$ (via supervised learning)
8:     Adjust $\epsilon$ (see [16, §4.2])
9: **end for**

---

# 5 TWO-PLAYER ZERO-SUM ITERATIVE DYNAMIC GPS

To guarantee robust performance during the training of policies of a stochastic system, we introduce the "worst" disturbance in the $H_\infty$ paradigm to the search for a good guiding distribution problem. We begin by augmenting the reward function with a term that allows for withstanding a disturbing input

$$\ell(\mathbf{x}_t, \mathbf{u}_t, \mathbf{v}_t, t) = r(\mathbf{x}_t, \mathbf{u}_t, \mathbf{v}_t, t) + \gamma^2 \mathbf{v}^T \mathbf{v}. \tag{6}$$

where $\gamma^2 \mathbf{v}^T \mathbf{v}$ allows us to introduce a quadratic weighting term in the disturbing input; $\gamma$ denotes the robustness parameter. A zero-sum game follows explicitly: the protagonist is guided toward regions of high reward regions while adversary pulls in its own favorite direction – yielding a *saddle-point* solution. This framework facilitates learning control decision strategies that are robust in the presence of disturbances and modeling errors – improving upon the generic optimal control policies that GPS and indeed deep RL algorithms guarantee.

## 5.1 TWO-PLAYER TRAJECTORY OPTIMIZATION

We propose repeatedly solving an MPC-based finite-horizon trajectory optimization problem within the framework of DDP. Specifically, we generalize a DDP variant – the iLQG algorithm of [22], to a two-player, zero-sum dynamic game as follows:

- we iteratively approximate the nonlinear dynamics, $\dot{\mathbf{x}} = f(\mathbf{x}_t, \mathbf{u}_t, \mathbf{v}_t)$, starting with nominal control, $\bar{\mathbf{u}}_t; t \in [t_0, t_f]$, and nominal adversarial input $\bar{\mathbf{v}}_t; t \in [t_0, t_f]$ which are assumed to be available.

- we run the passive dynamics with $\bar{\mathbf{u}}_t$ and $\bar{\mathbf{v}}_t$ to generate a trajectory $(\bar{\mathbf{x}}_t, \bar{\mathbf{u}}_t, \bar{\mathbf{v}}_t)$

- discretizing time, we linearize the nonlinear system, $\dot{\mathbf{x}}_t$, about $(\bar{\mathbf{x}}_k, \bar{\mathbf{u}}_k, \bar{\mathbf{v}}_k)$, so that the new state and action pairs become

$$\delta\mathbf{x}_k = \mathbf{x}_k - \bar{\mathbf{x}}_k, \quad \delta\mathbf{u}_k = \mathbf{u}_k - \bar{\mathbf{u}}_k, \quad \delta\mathbf{v}_k = \mathbf{v}_k - \bar{\mathbf{v}}_k$$

$\delta\mathbf{x}_k, \delta\mathbf{u}_k$, and $\delta\mathbf{v}_k$ are measured w.r.t the nominal vectors $\bar{\mathbf{x}}_k, \bar{\mathbf{u}}_k, \bar{\mathbf{v}}_k$ and are not necessarily small. The LQG approximation to the original optimal control problem and reward become

$$\delta\mathbf{x}_{k+1} \approx f_{\mathbf{X}k}\delta\mathbf{x}_k + f_{\mathbf{U}k}\delta\mathbf{u}_k + f_{\mathbf{V}k}\delta\mathbf{v}_k$$

$$\ell(\mathbf{x}_k, \mathbf{u}_k, \mathbf{v}_k) \approx \delta\mathbf{x}_k^T \ell_{\mathbf{x}k} + \delta\mathbf{u}_k^T \ell_{\mathbf{u}k} - \gamma\delta\mathbf{v}_k^T \ell_{\mathbf{v}k} + \frac{1}{2}\delta\mathbf{x}_k^T \ell_{\mathbf{xx}k}\delta\mathbf{x} + \frac{1}{2}\delta\mathbf{u}_k^T \ell_{\mathbf{uu}k}\delta\mathbf{u} + \frac{1}{2}\gamma^2 \delta\mathbf{v}_k^T \ell_{\mathbf{vv}k}\delta\mathbf{v}$$
$$+ \delta\mathbf{u}\ell_{\mathbf{u}^T\mathbf{X}k}\delta\mathbf{x} - \gamma\delta\mathbf{v}^T \ell_{\mathbf{vx}k}\delta\mathbf{x} + \ell(\bar{\mathbf{x}}_k, \bar{\mathbf{u}}_k, \bar{\mathbf{v}}_k) + \mathbb{E}(\mathbf{w}_t).$$

where single and double subscripts in the augmented reward denote first-order and second-order derivatives respectively, and $f_{\mathbf{z}k}$ are the respective Jacobians e.g. $f_{\mathbf{X}k} = \frac{\partial f(\cdot)}{\partial \mathbf{X}}|_k$ and $f_{\mathbf{xx}k} = \frac{\partial}{\partial \mathbf{X}}\frac{\partial f(\cdot)}{\partial \mathbf{X}}|_k$ at time $k$, $\mathbb{E}(\mathbf{w}_t)$ is an additive random noise term (folded into $\mathbf{v}_t$ in our implementation); the value function is the cost-to-go given by the min-max of the control sequence

$$V(\mathbf{x}_k) = \min_{\mathcal{U}_i} \max_{\mathcal{V}_i} \ell_{i,j}(\mathbf{x}_k, \mathcal{U}_i, \mathcal{V}_j).$$

Setting $V(\mathbf{x}_{k_f}) = \ell_{k_f}(\mathbf{x}_{k_f})$, where $k_f$ is the final time step, the dynamic programming problem transforms the min-max over an entire control sequence to a series of optimizations over a single control, which proceeds backward in time as

$$V(\mathbf{x}_k) = \min_{p\mathbf{u}_k} \max_{p\mathbf{v}_k} [\ell(\mathbf{x}_k, \mathbf{u}_k, \mathbf{v}_k) + V(f(\mathbf{x}_{k+1}, \mathbf{u}_{k+1}, \mathbf{v}_{k+1}))].$$

The Hamiltonian, $\ell(\cdot) + V(\cdot)$, can be considered as a function of perturbations around the tuple $\{\mathbf{x}_k, \mathbf{u}_k, \mathbf{v}_k\}$. Given the intractability of solving the Bellman partial differential equation above, we restrict our attention to the local neighborhood of the nominal trajectory by expanding a power series about the nominal, nonoptimal trajectory similar to [18]. We proceed as follows:

- we maintain a second-order local model of the perturbed $Q$-coefficients of the LQR problem, $(Q_k, Q_{\mathbf{x}k}, Q_{\mathbf{u}k}, Q_{\mathbf{v}k}, Q_{\mathbf{xx}k}, Q_{\mathbf{ux}k}, Q_{\mathbf{vx}k}, Q_{\mathbf{uu}k}, Q_{\mathbf{vv}k})^2$, defined thus

$$Q(\delta\mathbf{x}_k, \delta\mathbf{u}_k, \delta\mathbf{v}_k, k) = \ell(\mathbf{x}_k + \delta\mathbf{x}_k, \mathbf{u}_k + \delta\mathbf{u}_k, \mathbf{v}_k + \delta\mathbf{v}_k) - \ell(\mathbf{x}_k, \mathbf{u}_k\mathbf{v}_k) - V(f(\mathbf{x}_k, \mathbf{u}_k\mathbf{v}_k))$$
$$+ V(f(\mathbf{x}_k + \delta\mathbf{x}_k, \mathbf{u}_k + \delta\mathbf{u}_k, \mathbf{v}_k + \delta\mathbf{v}_k)),$$

- a second-order Taylor approximation of $Q(\delta\mathbf{x}_k, \delta\mathbf{u}_k, \delta\mathbf{v}_k, k)$ in the preceding equation yields

$$\approx \frac{1}{2} \begin{bmatrix} 1 \\ \delta\mathbf{x}_k^T \\ \delta\mathbf{u}_k^T \\ \delta\mathbf{v}_k^T \end{bmatrix}^T \begin{bmatrix} 0 & Q_{\mathbf{x}k}^T & Q_{\mathbf{u}k}^T & Q_{\mathbf{v}k}^T \\ Q_{\mathbf{x}k} & Q_{\mathbf{xx}k} & Q_{\mathbf{xu}k} & Q_{\mathbf{xv}k} \\ Q_{\mathbf{u}k} & Q_{\mathbf{ux}k} & Q_{\mathbf{uu}k} & Q_{\mathbf{uv}k} \\ Q_{\mathbf{v}k} & Q_{\mathbf{vx}k} & Q_{\mathbf{vu}k} & Q_{\mathbf{vv}k} \end{bmatrix} \begin{bmatrix} 1 \\ \delta\mathbf{x}_k \\ \delta\mathbf{u}_k \\ \delta\mathbf{v}_k \end{bmatrix} \tag{7}$$

- the best possible (protagonist) action and the worst possible (adversarial) action can be found by performing the respective arg min and arg max operations

$$\delta\mathbf{u}_k^\star = \arg\min_{\delta\mathbf{u}_k} Q(\delta\mathbf{x}_k, \delta\mathbf{u}_k, \delta\mathbf{v}_k), \text{ and } \delta\mathbf{v}_k^\star = \arg\max_{\delta\mathbf{v}_k} Q(\delta\mathbf{x}_k, \delta\mathbf{u}_k, \delta\mathbf{v}_k)$$

so that we have the following linear controllers that minimize and maximize the quadratic Q-function respectively:

$$\delta\mathbf{u}_k^\star = -Q_{\mathbf{uu}k}^{-1} \left[ Q_{\mathbf{u}k}^T + Q_{\mathbf{ux}k}\delta\mathbf{x}_k + Q_{\mathbf{uv}k}\delta\mathbf{v}_k \right], \quad \delta\mathbf{v}_k^\star = -Q_{\mathbf{vv}k}^{-1} \left[ Q_{\mathbf{v}k}^T + Q_{\mathbf{vx}k}\delta\mathbf{x}_k + Q_{\mathbf{vu}k}\delta\mathbf{u}_k \right].$$

For nonlinear systems, the inverse of the 2nd partial derivatives of the Hamiltonian with respect to the controls must be strictly positive definite. When the inverse of the Hessians above are non-positive-definite, we can circumvent this bottleneck by adding a suitably large positive quantity to $Q_{\mathbf{uu}k}^{-1}$ and $Q_{\mathbf{vv}k}^{-1}$ [12, 5], by replacing the Hessian with an identity matrix (which gives the steepest descent) [23], or by multiplying by lowest eigenvalue of the matrix. We find that the protagonist and adversary in the above-equations have a local action containing a state feedback term, $\mathbf{G}$, and an open-loop term, $\mathbf{g}$, given by

$$\mathbf{g}_{\mathbf{u}_k} = -Q_{\mathbf{uu}k}^{-1}[Q_{\mathbf{u}k} + Q_{\mathbf{uv}k}\delta\mathbf{v}_k], \ \mathbf{G}_{\mathbf{u}_k} = -Q_{\mathbf{uu}k}^{-1}Q_{\mathbf{ux}k},$$
$$\mathbf{g}_{\mathbf{v}_k} = -Q_{\mathbf{vv}k}^{-1}[Q_{\mathbf{v}k} + Q_{\mathbf{vu}k}\delta\mathbf{u}_k], \ \mathbf{G}_{\mathbf{v}_k} = -Q_{\mathbf{vv}k}^{-1}Q_{\mathbf{vx}k}. \tag{8}$$

respectively. The tuple $\{\mathbf{g}_{\mathbf{u}_k}, \mathbf{G}_{\mathbf{u}_k}, \mathbf{g}_{\mathbf{v}_k}, \mathbf{G}_{\mathbf{v}_k}\}$ can be computed efficiently as shown in (17). We can construct linear Gaussian controllers with mean given by the deterministic optimal solutions and the covariance proportional to the curvatures of the respective Q functions:

$$p(\mathbf{u}_k|\mathbf{x}_k) = \mathcal{N}(\bar{\mathbf{u}} + \mathbf{g}_{\mathbf{u}k} + \mathbf{G}_{\mathbf{u}k}\delta\mathbf{x}_k, Q_{\mathbf{uu}k}^{-1}),$$

$$p(\mathbf{v}_k|\mathbf{x}_k) = \mathcal{N}(\bar{\mathbf{v}} + \mathbf{g}_{\mathbf{v}k} + \mathbf{G}_{\mathbf{v}k}\delta\mathbf{x}_k, Q_{\mathbf{vv}k}^{-1}).$$

[13] has shown that these types of distributions optimize an objective function with maximum entropy given by

$$\arg\min_{p(\bar{\tau})\in\mathcal{N}(\bar{\tau})} \mathbb{E}[\ell(\bar{\tau}) - \mathcal{H}(p(\bar{\tau}))] \quad \text{subject to} \quad p(\mathbf{x}_{t+1}|\mathbf{x}_t, \mathbf{u}_t) = \mathcal{N}(\mathbf{x}_{t+1}; f_{\mathbf{x}t}\mathbf{x}_t + f_{\mathbf{u}t}\mathbf{u}_t, \mathbf{F}_t) \tag{9}$$

---

[2]where $Q_k = \ell(\mathbf{x}_k, \mathbf{u}_k, \mathbf{v}_k, k) + V(\mathbf{x}_{k+1}, k+1)$. Vector subscripts indicate partial derivatives.

while $p(\mathbf{v}_k|\mathbf{x}_k)$ optimizes

$$\arg \max_{p(\bar{\tau}) \in \mathcal{N}(\bar{\tau})} \mathbb{E}[\ell(\bar{\tau}) - \mathcal{H}(p(\bar{\tau}))] \quad \text{subject to} \quad p(\mathbf{x}_{t+1}|\mathbf{x}_t, \mathbf{v}_t) = \mathcal{N}(\mathbf{x}_{t+1}; f_{\mathbf{x}t}\mathbf{x}_t + f_{\mathbf{v}t}\mathbf{v}_t, \mathbf{F}_t)$$
(10)

where $\bar{\tau} = (\mathbf{x}_t^i, \mathbf{u}_t^i, \mathbf{v}_t^i)$ is the system's trajectory evolution over all states, $i$, visited by both local controllers, and $\mathcal{H}$ is the differential entropy. Equation (9) produces a trajectory that follows the widest, highest-entropy distribution while minimizing the expected cost under linearized dynamics and quadratic cost; (10) produces an opposing trajectory to what $p(\mathbf{u}_k|\mathbf{x}_k)$ does by maximizing the expected cost under locally linear quadratic assumptions about the dynamics.

Note that the open-loop control strategies in (8) depend on the action of the other player. Therefore, equations (8) ensure we have a *cooperative game* in which the protagonist and the adversary alternate between taking best possible and worst possible local actions during the trajectory optimization phase. This helps maintain equilibrium around the system's desired trajectory, while ensuring robustness in local policies. Substituting (8) into (7) and equating coefficients of $\delta\mathbf{x}_k, \delta\mathbf{u}_k, \delta\mathbf{v}_k$ to those of $V(\mathbf{x}_k + \delta\mathbf{x}_k, k) = V(\mathbf{x}_k, k) + V_{\mathbf{X}k}\delta\mathbf{x}_k + \frac{1}{2}\delta\mathbf{x}_k^T V_{\mathbf{XX}k}\delta\mathbf{x}_k$, we obtain a quadratic value function at time $k$, through the backward pass given by (19) in the appendix.

Say, the protagonist first implements its strategy, then transmits its information to the adversary, who subsequently chooses its strategy; it follows that the adversary can choose a more favorable outcome since it knows what the protagonist's choice of strategy is. It becomes obvious that the *best* action for the protagonist is to choose a control strategy that is an optimal response to the choice of the adversary determined from

$$\delta\mathbf{v}_k = \min_{p_{\delta\mathbf{v}_k}} \ell(\delta\mathbf{x}_k, \delta\mathbf{u}_k, \delta\mathbf{v}_k) = \max_{p_{\delta\mathbf{v}_k}} \min_{p_{\delta\mathbf{u}_k}} \ell(\delta\mathbf{x}_k, \delta\mathbf{u}_k, \delta\mathbf{v}_k).$$

Similarly, if the roles of the players are changed, the protagonist response to the adversary's *worst* choice will be

$$\delta\mathbf{u}_k = \max_{\delta\mathbf{u}_k} \ell(\delta\mathbf{x}_k, \delta\mathbf{u}_k, \delta\mathbf{v}_k) = \min_{p_{\delta\mathbf{u}_k}} \max_{p_{\mathbf{v}_k}} \ell(\delta\mathbf{x}_k, \delta\mathbf{u}_k, \delta\mathbf{v}_k).$$

Therefore, it does not matter that the order of play is predetermined. We end up with an *iterative dynamic game*, where each agent's strategy depends on its previous actions. The update rules for the $Q$ coefficients are determined using a Gauss-Newton approximation and is given in (15) in the appendices.

In the forward pass, we integrate the state equation, $\dot{\mathbf{x}}$, compute the protagonist's deterministic optimal policy and update the trajectory as follows:

$$\bar{\mathbf{g}}(\mathbf{x}_k) = \bar{\mathbf{u}}_k + \mathbf{g}_{\mathbf{u}k} + \mathbf{G}_{\mathbf{u}k}(\mathbf{x}_k - \bar{\mathbf{x}}_k)$$
$$\mathbf{x}_1 = \bar{\mathbf{x}}_1, \quad \bar{\mathbf{x}}_{k+1} = f(\bar{\mathbf{x}}_k, \bar{\mathbf{u}}_k, \bar{\mathbf{v}}_k)$$
(11)

Compared to previous GPS algorithms, the local controllers not only produce locally linear Gaussian controllers that favors the widest and highest entropy, they also have robustness to disturbance and modeling errors built into them in the $H_\infty$ sense.

We arrive at a saddle point in the energy space of the cost function and we posit that the local controllers generated during the trajectory optimization phase become robust to external perturbations, modeling errors e.t.c. We arrive at a saddle point in the energy space of the cost function and we posit that the local controllers generated during the trajectory optimization phase become robust to external perturbations, modeling errors e.t.c. The next section shows how we generate the function $V(\mathbf{x}_k)$ that guarantees saddle-point equilibria for our examples.

## 5.2 ESTIMATING DYNAMICS DISTRIBUTION

The dynamics of the two player system is given by the tuple $\{\mathbf{x}_t^i, \mathbf{u}_t^i, \mathbf{v}_t^i, \mathbf{x}_{t+1}^i\}$ and we fit the system dynamics using piecewise linear functions in the form of a mixture of $N$ Gaussians as proposed in [1] and [14] over the vectors $\{\mathbf{x}_t^i, \mathbf{u}_t^i, \mathbf{v}_t^i, \mathbf{x}_{t+1}^i\}^T$, where the $i$th index represents the $i$-th trajectory rollout on the robot. We build a Gaussian Mixture Model (GMM) to fit piecewise linear dynamics so that within each GMM cluster, $k_i$, we represent a linear Gaussian dynamics model as $k_i(\mathbf{x}_{t+1}^i|\mathbf{x}_t^i, \mathbf{u}_t^i, \mathbf{v}_t^i)$

and the marginal $k_i(\mathbf{x}_t^i | \mathbf{u}_t^i, \mathbf{v}_t^i)$ represents the portion of the state-actions space where our Gaussian model is valid.

In order to avoid the GMM not being a good separator of boundaries of complex modes, we follow [1], and use the GMM to generate a prior for the regression phase. This enables us to obtain different linear modes at separate time steps based on the observed transitions, even when the states are dissimilar. The correct linear mode is obtained from the empirical covariance of $\{\mathbf{x}_t, \mathbf{u}_t, \mathbf{v}_t\}$ with $\mathbf{x}_{t+1}$ in the current samples at time $t$. As in [1] and [14], we improve sample efficiency by refitting the GMM at each iteration to all of the samples at all time steps from the current iteration and the previous 3 iterations and use this to construct a good prior for the dynamics. We then obtain linear Gaussian dynamics by fitting Gaussian distributions to samples $\{\mathbf{x}_t^i, \mathbf{u}_t^i, \mathbf{v}_t^i, \mathbf{x}_{t+1}^i\}$ which are then conditioned on $[\mathbf{x}_t, \mathbf{u}_t, \mathbf{v}_t]^T$. The prior allows us to build a normal-inverse Wishart prior on the conditioned Gaussians so that the maximum a posteriori estimates for mean $\mu$ and covariance $\Sigma$ are given by

$$\Sigma = \frac{\Phi + N\Sigma_e + \frac{Nm}{N+m}(\mu_e - \mu_0)(\mu_e - \mu_0)^T}{N + n_0}, \quad \mu = \frac{m\mu_0 + n_0\mu_e}{m + n_0}$$

where $\Sigma_e$ and $\mu_e$ are respectively the empirical covariance and mean of the dataset and $\Phi, \mu_0, m$ and $n_0$ are prior parameters so chosen: $\Phi = n_0\bar{\Sigma}$ and $\mu_0 = \bar{\mu}$. As in [14], we set $n_0 = m = 1$ in order to fit the prior to many samples than what is available at each time step.

## 5.3 SUPERVISED LEARNING OF GLOBAL NEURAL NETWORK POLICIES

The trajectories from the previous subsection are used to generate training data for global policies for the controller and adversary. The local policies $p_{\bar{\tau}}(\mathbf{u}_k|\mathbf{x}_k)$ and $p_{\bar{\tau}}(\mathbf{v}_k|\mathbf{x}_k)$ will ideally be generated for all possible initial states $\mathbf{x}_1^i \sim p(\mathbf{x}_k)$. Since the iLQG-based linearized dynamics will only be valid within a finite region of the state space; we used the KL-divergence constraint proposed in [16] to ensure the current protagonist policy does not diverge too much from the previous policy.

The learning problem involves imposing KL constraints on the cost function such that the protagonist controller distribution agree with the global policy $\pi_\theta(\mathbf{u}_t|\mathbf{x}_t)$ by performing the following alternating optimization between two steps at each iteration $i$:

$$\arg \min_{p\mathbf{u}_i} \max_{p\mathbf{v}_i} \left[ \mathbb{E}_p(\bar{\tau}) \sum_{k=1}^K l(\mathbf{x}_k, \mathbf{u}_k, \mathbf{v}_k) \right] \text{ s.t. } D_{KL}(p_i(\mathbf{u}_k|\mathbf{x}_k), \pi_i) \leq \epsilon,$$
$$\pi_{i+1} \leftarrow \arg \min_{\pi \in \Pi_\Theta} D_{KL}(p_i(\mathbf{u}_k|\mathbf{x}_k), \pi), \tag{12}$$

Essentially, $p(\mathbf{u}_k|\mathbf{x}_k)$ above generates robust local policies; The first step in (12) solves for a robust local policy $p(\mathbf{u}_k|\mathbf{x}_k)$ via the min-max operation, by constraining $p(\mathbf{u}_k|\mathbf{x}_k)$ against its global policy $\pi_i$ using the given KL divergence constraint; the second step projects the local linear Gaussian controller distribution onto the constraint set $\Pi_\Theta$, with respect to the divergence $D(p_i, \pi)$. The local policy that governs the agent's dynamics is given by

$$p(\mathbf{u}_k|\mathbf{x}_k) = \mathcal{N}(\bar{\mathbf{u}} + \mathbf{g}_{\mathbf{u}k} + \mathbf{G}_{\mathbf{u}k}\delta\mathbf{x}_k, Q_{\mathbf{u}\mathbf{u}k}^{-1}). \tag{13}$$

Notice that the state is linearly dependent on the mean of the distribution $p(\mathbf{u}_k|\mathbf{x}_k)$ and the covariance is independent of $\mathbf{v}_k$; we therefore end up with a linear Gaussian controller for the robust guided policy search algorithm. For linear Gaussian dynamics and policies, the iterative KL constraint during the S-step translates to minimizing the KL-divergence between policies $i.e.$ ,

$$D_{KL}(p_i(\bar{\tau}) \| \pi_\theta(\bar{\tau})) = \sum_{k=1}^K \mathbb{E}_{p(\mathbf{u}_k|\mathbf{x}_k)} D_{KL}(p(\mathbf{u}_k|\mathbf{x}_k) \| \pi_\theta(\mathbf{u}_k|\mathbf{x}_k)).$$

For the nonlinear cases that we treat in this work, the KL-divergence term in the S-step above is flipped as proposed in [16] so that $D_{KL}(\pi_\theta(\mathbf{u}_k|\mathbf{x}_k) \| p_i(\mathbf{u}_k|\mathbf{x}_k))$ minimizes the augmented stage cost under $p_i(\mathbf{u}_k|\mathbf{x}_k)$ w.r.t $\pi_\theta(\mathbf{u}_k|\mathbf{x}_k)$. Therefore, the S-step minimizes,

$$\sum_{i,k} E_{p_i(\mathbf{x}_k)} \left[ D_{KL}(\pi_\theta(\mathbf{u}_k|\mathbf{x}_k) \| p_i(\mathbf{u}_k|\mathbf{x}_k)) \right] \approx \frac{1}{|\mathcal{D}_i|} \sum_{i,k,j} D_{KL}(\pi_\theta(\mathbf{u}_{k,i,j}|\mathbf{x}_k) \| p_i(\mathbf{u}_{k,i,j}|\mathbf{x}_k)),$$

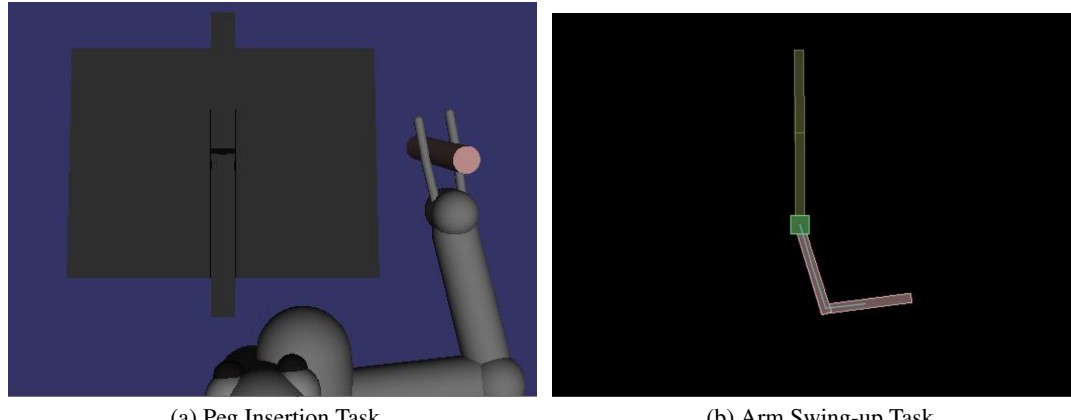

(a) Peg Insertion Task            (b) Arm Swing-up Task

Figure 1: Simulation Experiments

where $\mathbf{x}_{k,i,j}$ is the $j^{\text{th}}$ sample from $p_i(\mathbf{x}_k)$ obtained by running $p_i(\mathbf{u}_k|\mathbf{x}_k)$ on the real system, and $\mathcal{D}_i$ are the trajectory samples rolled out on the system. Our robust GPS algorithm is thus given in algorithm 2. We follow the prior works in [1, 13, 14, 16] in computing the KL divergence term and we refer readers to these works for a more detailed treatment.

## 6 EXPERIMENTAL RESULTS

In this section, we present experiments to (i) confirm our hypothesis that guided policy search methods, with carefully engineered complex high-dimensional policies, fail when exposed to the simplest of all perturbation signals; and (ii) answer the question of robustness using the trajectory optimization scheme and the robust guided policy framework we have presented. We solve this under unknown dynamics.

We answer both questions in this paper by using physics engines for policies that do not use visual features as feedback. Our validation examples are implemented in the *MuJoCo* physics engine [24] and the *pybox2d* game engine [6], aided by the publicly available GPS codebase [8]. High-dimensional policy experiments are implemented on a PR2 robot, while low-dimensional policy experiments are implemented using a 2-DOF cart-pole swing-up experiment in the *pybox2d* game engine. The perturbation signal we consider are those that enter additively through the reward functions as described in subsection 4.1.

### 6.1 SENSITIVITY OF AN RL POLICY

We conducted simulated experiments demonstrating that guided policy search policies are sensitive to disturbance introduced into the action space of their policies. The 7-DoF robot result presented shortly previously appeared in our abstract that introduced robust GPS [21]. The states $\mathbf{x}_k$ are the joint angles, joint velocities, pose and velocity of the end effector as 3 points in 3-D. We assume the initial velocity of the 2-link and robot arm are zero.

**Experimental tasks**. We simulated a 3D peg insertion task by a robot into a hole at the bottom of the slot Fig. 1. The difficulty of this experiment stems from the discontinuity in dynamics from the contact between the peg and the walls.

The 2-link arm swing-up experiment involves learning to balance the arm vertically about the origin of the cart (see right inset of Fig. 1). The diffculty lies in the discontinuity of the dynamics along the vertical axis of the arm when it is upright.

We initialized the linear-Gaussian controllers $p_i(\mathbf{u}_k|\mathbf{x}_k)$, $p_i(\mathbf{v}_k|\mathbf{x}_k)$ in the neighborhood of the initial state $\mathbf{x}_1$ using a PD control law for both the inverted pendulum task and the peg insertion task.

**Peg Insertion:** We implement the sensitivity algorithm for the peg insertion task of [8] with a robotic arm that requires dexterous manipulation. The robot has 12 states consisting of joint angles

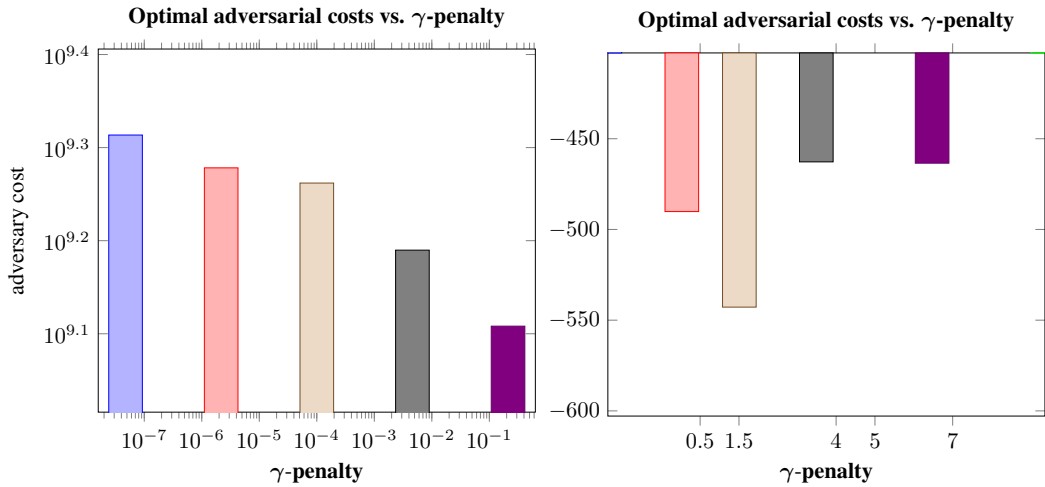

Figure 2: Sensitivity Analysis for Peg Insertion Task

and angular velocities with two controller states. We train the protagonist's policy using the GPS algorithm. We then pit an adversarial disturbance against the trained policy so that the adversary stays in closed-loop with the trained protagonist; The closed-loop cost function is given by

$$\ell(\mathbf{x}_k, \mathbf{u}_k, \mathbf{v}_k) = \frac{1}{2} w_{\mathbf{u}} \mathbf{u}_k^T \mathbf{u}_k + w_{\mathbf{p}} \ell_{12}(\mathbf{d}_{\mathbf{x}_k} - \mathbf{d}^\star) - \gamma \mathbf{v}_k^T \mathbf{v}_k \tag{14}$$

where $\gamma$ represents the disturbance term, $\mathbf{d}_{\mathbf{x}_k}$ denotes the end effector's (EE) position at state $\mathbf{x}_k$ and $\mathbf{d}^\star$ denotes the EE's position at the slot's base. $\ell_{12}(\zeta)$ is a term that makes the peg reach the target at the hole's base, precisely given by $\frac{1}{2}\zeta^t\zeta + (\alpha + \zeta^2)^{\frac{1}{2}}$. We set $w_{\mathbf{u}}$ and $w_{\mathbf{p}}$ to $10^{-6}$ and 1 respectively. For various values of $\gamma$, we check the sensitivity of the trained policy and its effect on the task performance by maximizing the cost function above w.r.t $\mathbf{v}_k$. We run each sensitivity experiment for a total of 10 iterations. Fig. 2 shows that the adversary causes a sharp degradation in the protagonist's performance for values of $\gamma < 1.5$. This corresponds to when the GPS-trained policy gets destabilized and the arm struggles to reach the desired target. As values of $\gamma \geq 1.5$, however, we find that the adversary has a reduced effect on task performance: the adversary's effect decreases as $\gamma$ gets larger. Video of this result is available at https://goo.gl/YmmdhC.

**Arm Swing-up**: Similar to the peg insertion task, we carry out a sensitivity evaluation procedure as we did for the robot arm with the peg insertion experiment with a 2D arm. The goal is to balance a 2D arm vertically about its origin. This agent has 7 states made up of two joint angles, two joint angle velocities and a 3D end effector point. The action space has two dimensions. Contrary to the example in [8] that uses the Bregman alternating direction method of multipliers algorithm, we implement this experiment using the mirror descent GPS algorithm. We proceed as before: first, we optimize the optimal global policy using GPS on the agent; we then fix the agent's policy and pit various adversarial disturbances, controlled by the $\gamma$ robustness term in order to evaluate its sensitivity. We use a similar cost function as the one used for the peg insertion task. Fig. 3 shows the evolution of the cost function as we vary the values of $\gamma$. We notice that the augmented reward function gets larger as the adversary's torque increases in magnitude and for lower values of $\gamma$, the augmented cost is relatively low stays the same. The values of $\gamma < 10^{12}$ in Fig. 3 represent the disturbance band where the protagonist's learned policy becomes unstable and the arm never reaches the vertical position (see videos here: https://goo.gl/52rKnt). This experiment further confirms that the state-of-the-art reinforcement learning algorithms fail in the presence of additive disturbances to their parameter space making them brittle when used in situations that call for robustness. To mitigate these sensitivity errors, we implement the robust two-player, zero-sum game framework provided in 5 in order to develop more robust deep RL controllers and mitigate modeling errors and uncertainty.

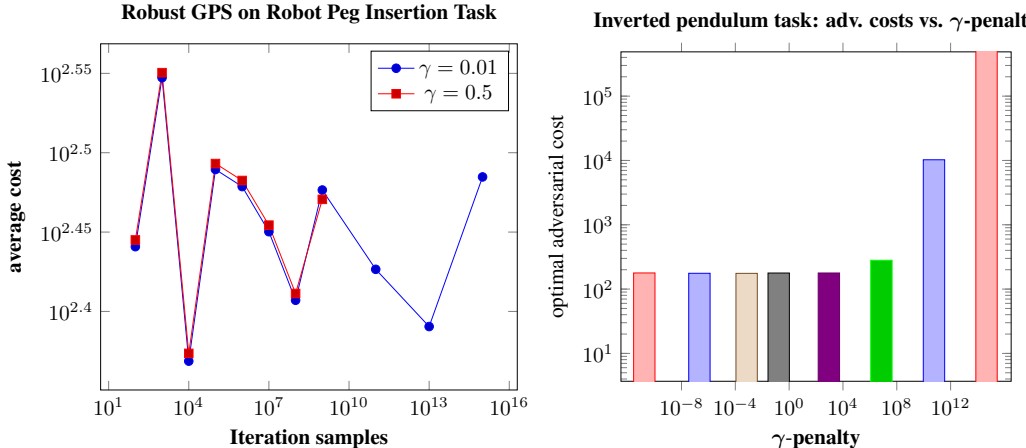

Figure 3: [LEFT]: Cost of running the dynamic game-based robust guided policy search algoruithm for various values of gamma for the robot peg insertion task. Our algorithm uses lesser number for the Gaussian mixture models and requires fewer samples to generalize to the real-world. RIGHT: Sensitivity Analysis for Arm Swing-up Task

## 6.2 ROBUST RL WITH GPS

As proposed in section 5, our goal is to improve the robustness of the controller's policy in the presence of modeling errors and uncertainties and transfer errors. We follow the formulation in section 5 and generate $\mathbf{v}_k$ from zero-mean, unit variance noise samples in every iteration. We employ various values of $\gamma$ as a robustness parameter and we run the dynamic game during the trajectory optimization phase of the GPS algorithm. Specifically, for the values of $\gamma$ that the erstwhile policies in the previous subsection fail, we run the dynamic game algorithm to provide robustness in performnace at test time compared against the GPS algorithm. We run experiments on the peg insertion task to verify the algorithm. Figure 3 shows the cost of running the robust GPS algorithm on the the 7-DoF robot. We see that the policies that show achieve optimal performance behavior are now less costly compared to vanilla GPS algorithm. For values of the sensitivity term $\gamma$ that the algorithm erstwhile fails in, we now see smoother execution of the trajectory in trying to achieve our goal. The modeling phase of the algorithm is also much less data consuming as our GMM algorithm now takes less samples before generalizing to the global model.

## 7 CONCLUSION AND FUTURE WORK

We have evaluated the sensitivity of select deep reinforcement learning algorithms and shown that despite the most carefully designed policies, such policies implemented on real-world agents exhibit a potential for disastrous performances when unexpected such as when there exist modeling errors and discrepancy between training environment and real-world roll-outs (as evidenced by the results from the two dynamics the agent faces in our sensitivity experiment). We then test the dynamic trajectory optimization two-player algorithm on a robot motor task using Levine et al's [14]'s guided policy search algorithm. In our implementation of the dynamic game algorithm, we focus on the robustness parameters that cause the robot's policy to fail in the presence of the erstwhile *sensy*-sensitivity parameter. We demonstrate that our two-player game framework allows agents operating under nonlinear dynamics to learn the underlying dynamics under significantly more finite samples than vanilla GPS algorithm does – thus improving upon the Gaussian model mixture method used in [1] and [14].

Having agents that are robust to unmodeled nonlinearities, dynamics, and high frequency modes in a nonlinear dynamical system has long been a fundamental question that control theory strives to achieve. To the best of our knowledge, we are not aware of other works that addresses the robustness of deep policies that are trained end-to-end from a maximal robustness perspective. In future work, we hope to replace the crude Gaussian Mixture Model for the dynamics with a more sophisticated nonlinear model, and evaluate how the agent behaves in the presence of unknown dynamics.

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

## APPENDIX I

The Q-coefficients are estimated using LQR as follows:

$$
\begin{aligned}
&Q_{\mathbf{x}k} = \ell_{\mathbf{x}k} + f_{\mathbf{X}k}^T V_{\mathbf{X}k+1}, &&Q_{\mathbf{u}k} = \ell_{\mathbf{u}k} + f_{\mathbf{u}k}^T V_{\mathbf{X}k+1} \\
&Q_{\mathbf{v}k} = \ell_{\mathbf{v}k} + f_{\mathbf{V}k}^T V_{\mathbf{X}k+1} &&Q_{\mathbf{XX}k} = \ell_{\mathbf{xx}k} + f_{\mathbf{X}k}^T V_{\mathbf{XX}k+1} f_{\mathbf{X}k} \\
&Q_{\mathbf{ux}k} = \ell_{\mathbf{ux}k} + f_{\mathbf{u}k}^T V_{\mathbf{XX}k+1} f_{\mathbf{X}k} &&Q_{\mathbf{vx}k} = \ell_{\mathbf{vx}k} + f_{\mathbf{V}k}^T V_{\mathbf{XX}k+1} f_{\mathbf{X}k} \\
&Q_{\mathbf{uu}k} = \ell_{\mathbf{uu}k} + f_{\mathbf{u}k}^T V_{\mathbf{XX}k+1} f_{\mathbf{u}k} &&Q_{\mathbf{vv}k} = \ell_{\mathbf{vv}k} + f_{\mathbf{V}k}^T V_{\mathbf{XX}k+1} f_{\mathbf{V}k},
\end{aligned}
\tag{15}
$$

where the subscript terms denote the partial derivatives with respect to the given matrix.

## APPENDIX II

The closed-form equations for the closed loop policy obtained after substituting (8) are given by

$$
\delta \mathbf{u}_k^\star = -Q_{\mathbf{uu}k}^{-1} \left[ Q_{\mathbf{u}k}^T + Q_{\mathbf{ux}k}\delta\mathbf{x}_k + Q_{\mathbf{uv}k}\delta\mathbf{v}_k \right], \quad \delta \mathbf{v}_k^\star = -Q_{\mathbf{vv}k}^{-1} \left[ Q_{\mathbf{v}k}^T + Q_{\mathbf{vx}k}\delta\mathbf{x}_k + Q_{\mathbf{vu}k}\delta\mathbf{u}_k \right]. \tag{16}
$$

Solving the system of equations in (16), we find that

$$
\delta\mathbf{u}_k^\star = \left[ Q_{\mathbf{uu}k}\left( I - Q_{\mathbf{uu}k}^{-1}Q_{\mathbf{uv}k}Q_{\mathbf{vv}k}^{-1}Q_{\mathbf{uv}k}^T \right) \right]^{-1} \left[ \left( Q_{\mathbf{uv}k}Q_{\mathbf{vv}k}^{-1}Q_{\mathbf{vx}k} - Q_{\mathbf{ux}k} \right)\delta\mathbf{x}_k + \left( Q_{\mathbf{uv}k}Q_{\mathbf{vv}k}^{-1}Q_{\mathbf{v}k} - Q_{\mathbf{u}k}^T \right) \right]
\tag{17}
$$

$$
\delta\mathbf{v}_k^\star = \left[ Q_{\mathbf{uv}k}\left( I - Q_{\mathbf{vv}k}^{-1}Q_{\mathbf{vu}k}Q_{\mathbf{uu}k}^{-1} \right) Q_{\mathbf{vv}k} \right]^{-1} \left[ \left( Q_{\mathbf{vu}k}Q_{\mathbf{uu}k}^{-1}Q_{\mathbf{ux}k} - Q_{\mathbf{vx}k} \right)\delta\mathbf{x}_k + \left( Q_{\mathbf{vu}k}Q_{\mathbf{uu}k}^{-1}Q_{\mathbf{u}k}^T - Q_{\mathbf{v}k}^T \right) \right]
$$

Suppose we let

$$
\mathbf{K}_{\mathbf{u}k} = \left[ Q_{\mathbf{uu}k}\left( I - Q_{\mathbf{uu}k}^{-1}Q_{\mathbf{uv}k}Q_{\mathbf{vv}k}^{-1}Q_{\mathbf{uv}k}^T \right) \right]^{-1}, \quad \mathbf{K}_{\mathbf{v}k} = \left[ Q_{\mathbf{vv}k}\left( I - Q_{\mathbf{vv}k}^{-1}Q_{\mathbf{vu}k}Q_{\mathbf{uu}k}^{-1}Q_{\mathbf{uv}k} \right) \right]^{-1},
$$

so that

$$\mathbf{g}_{\mathbf{u}k} = \mathbf{K}_{\mathbf{u}k}(Q_{\mathbf{uv}k}Q_{\mathbf{vv}k}^{-1}Q_{\mathbf{v}k} - Q_{\mathbf{u}k}^T), \quad \mathbf{G}_{\mathbf{u}k} = \mathbf{K}_{\mathbf{u}k}\left(Q_{\mathbf{uv}k}Q_{\mathbf{vv}k}^{-1}Q_{\mathbf{vx}k} - Q_{\mathbf{ux}k}\right) \text{ and}$$
$$\mathbf{g}_{\mathbf{v}k} = \mathbf{K}_{\mathbf{v}k}(Q_{\mathbf{vu}k}Q_{\mathbf{uu}k}^{-1}Q_{\mathbf{u}k}^T - Q_{\mathbf{v}k}^T), \quad \mathbf{G}_{\mathbf{v}k} = \mathbf{K}_{\mathbf{v}k}\left(Q_{\mathbf{vu}k}Q_{\mathbf{uu}k}^{-1}Q_{\mathbf{ux}k} - Q_{\mathbf{vx}k}\right)$$

it follows that we can rewrite $\delta\mathbf{u}_k^\star$ and $\delta\mathbf{v}_k^\star$ as

$$\delta\mathbf{u}_k^\star = \mathbf{g}_{\mathbf{u}_k} + \mathbf{G}_{\mathbf{u}_k}\delta\mathbf{x}_k, \quad \delta\mathbf{v}_k^\star = \mathbf{g}_{\mathbf{v}_k} + \mathbf{G}_{\mathbf{v}_k}\delta\mathbf{x}_k. \tag{18}$$

Plugging $\delta\mathbf{u}^\star$ and $\delta\mathbf{v}^\star$ back into the Q function expansion in (7), we find that

$$
\begin{aligned}
Q(\delta\mathbf{x}_k, \delta\mathbf{u}_k, \delta\mathbf{v}_k, k) = {} & Q_{\mathbf{u}k}^T\mathbf{g}_{\mathbf{u}_k} + Q_{\mathbf{v}k}^T\mathbf{g}_{\mathbf{v}_k} + \frac{1}{2}\mathbf{g}_{\mathbf{u}k}^T Q_{\mathbf{uu}k}\mathbf{g}_{\mathbf{u}k} + \frac{1}{2}\mathbf{g}_{\mathbf{v}k}^T Q_{\mathbf{vv}k}\mathbf{g}_{\mathbf{v}k} \\
& (Q_{\mathbf{x}k}^T + Q_{\mathbf{u}k}^T\mathbf{G}_{\mathbf{u}k} + Q_{\mathbf{v}k}^T\mathbf{G}_{\mathbf{v}k} + \mathbf{g}_{\mathbf{u}k}^T Q_{\mathbf{uu}k}\mathbf{G}_{\mathbf{u}k} + \mathbf{g}_{\mathbf{v}k}^T Q_{\mathbf{vv}k}\mathbf{G}_{\mathbf{v}k} + \mathbf{g}_{\mathbf{u}k}^T Q_{\mathbf{ux}k} \\
& + \mathbf{g}_{\mathbf{v}k}^T Q_{\mathbf{vx}k} + \mathbf{g}_{\mathbf{u}k}^T Q_{\mathbf{uv}k}\mathbf{G}_{\mathbf{v}k} + \mathbf{g}_{\mathbf{v}k}^T Q_{\mathbf{uv}k}^T\mathbf{G}_{\mathbf{u}k})\delta\mathbf{x}_k \\
& + \frac{1}{2}\delta\mathbf{x}_k^T(Q_{\mathbf{xx}k} + \mathbf{G}_{\mathbf{u}_k}^T Q_{\mathbf{uu}k}\mathbf{G}_{\mathbf{u}} + \mathbf{G}_{\mathbf{v}}^T Q_{\mathbf{vv}k}\mathbf{G}_{\mathbf{v}} + 2\mathbf{G}_{\mathbf{u}_k}^T Q_{\mathbf{ux}k} \\
& + 2\mathbf{G}_{\mathbf{v}_k}^T Q_{\mathbf{vx}k} + 2\mathbf{G}_{\mathbf{u}_k}^T Q_{\mathbf{uv}k}\mathbf{G}_{\mathbf{v}_k})\delta\mathbf{x}_k
\end{aligned}
$$

Comparing coefficients, we obtain the following for the value function's coefficients

$$
\begin{aligned}
V_k - V_{k+1} = {} & Q_{\mathbf{u}k}^T\mathbf{g}_{\mathbf{u}k} + Q_{\mathbf{v}k}^T\mathbf{g}_{\mathbf{v}k} + \frac{1}{2}\mathbf{g}_{\mathbf{u}k}^T Q_{\mathbf{uu}k}\mathbf{g}_{\mathbf{u}k} + \frac{1}{2}\mathbf{g}_{\mathbf{v}k}^T Q_{\mathbf{vv}k}\mathbf{g}_{\mathbf{v}k} + \mathbf{g}_{\mathbf{u}k}^T Q_{\mathbf{uv}k}\mathbf{g}_{\mathbf{v}k} \\
V_{\mathbf{x}k} = {} & Q_{\mathbf{x}k}^T + Q_{\mathbf{u}k}^T\mathbf{G}_{\mathbf{u}_k} + Q_{\mathbf{v}k}^T\mathbf{G}_{\mathbf{v}_k} + \mathbf{g}_{\mathbf{u}k}^T Q_{\mathbf{uu}k}\mathbf{G}_{\mathbf{u}k} + \mathbf{g}_{\mathbf{v}k}^T Q_{\mathbf{vv}k}\mathbf{G}_{\mathbf{v}k} + \mathbf{g}_{\mathbf{u}k}^T Q_{\mathbf{ux}k} \\
& + \mathbf{g}_{\mathbf{v}k}^T Q_{\mathbf{vx}k} + \mathbf{g}_{\mathbf{u}k}^T Q_{\mathbf{uv}k}\mathbf{G}_{\mathbf{v}k} + \mathbf{g}_{\mathbf{v}k}^T Q_{\mathbf{uv}k}^T\mathbf{G}_{\mathbf{u}k} \\
V_{\mathbf{xx}k} = {} & Q_{\mathbf{xx}k} + \mathbf{G}_{\mathbf{u}_k}^T Q_{\mathbf{uu}k}\mathbf{G}_{\mathbf{u}k} + \mathbf{G}_{\mathbf{v}_k}^T Q_{\mathbf{vv}k}\mathbf{G}_{\mathbf{v}_k} + 2\mathbf{G}_{\mathbf{u}_k}^T Q_{\mathbf{ux}k} + 2\mathbf{G}_{\mathbf{v}_k}^T Q_{\mathbf{vx}k} + 2\mathbf{G}_{\mathbf{u}_k}^T Q_{\mathbf{uv}k}\mathbf{G}_{\mathbf{v}_k}
\end{aligned}
\tag{19}
$$

In practice, the $\mathbf{K}_{\mathbf{u}_k}$ and $\mathbf{K}_{\mathbf{v}_k}$ inverse terms above will result in numerical errors when the matrices are not positive definite since the DDP algorithm does not guarantee that the inverse of the Q functions will be positive definite. To guarantee numerical stability and preserve the concave-convex properties, we add to $\mathbf{K}_{\mathbf{u}_k}$ and $\mathbf{K}_{\mathbf{v}_k}$ a sufficiently large positive quantity (greater than the lowest eigenvalue) in order to regularize the update equations [12, 5].

