# OpenReview forum: "A dynamic game approach to training robust deep policies"
_ICLR.cc/2018/Conference — Reject_

### Official Review · AnonReviewer3 · 2017-11-25
**Interesting idea and good discussion, but unclear method and results**

**Rating:** 5
**Confidence:** 4

**Review:**

The authors propose to incorporate elements of robust control into guided policy search, in order to devise a method that is resilient to perturbations and (presumably) model mismatch.

The idea behind the method and the discussion in the introduction and related work is interesting and worthwhile, and I think that combining elements from robust control and reinforcement learning is a very promising direction to explore. However, in its present state, the paper is very hard to evaluate, perhaps because the submission was a bit rushed. It may be that the authors can clarify some of these issues in the response period.

First, the authors repeatedly state that perturbations are applied to the policy parameters. This seems very strange to me, as typically robust control considers perturbations to the state or control. And reading the actual method, I can't actually figure out how perturbations are applied to the parameters -- as near as I can tell, the perturbations are indeed applied to the controls. So which is it?

There is quite a lot of math in the derivation, and it's unclear which parts relate to the standard guided policy search algorithm, and which parts are new. After reading the technical sections several times, my best guess is that the method corresponds to using an adversarial trajectory optimization setup to generate supervision for training a policy. So only the trajectory optimization phase is actually different. Is that true? Or are there other modifications? Some sort of summary of the overall method would have been appreciated, or else a clearer separation of new and old components.

The evaluation also leaves a lot to be desired. What kind of perturbations are actually being considered? Are they all adversarial perturbations? Do the authors actually test model mismatch or other more natural conditions where robustness would be beneficial? In the end, I was unable to really interpret what the experiments are trying to get across, which makes it hard for me to tell if the method actually works or improves on anything.

In its present state, the paper is very hard to parse, and the evaluation appears too rushed for me to be able to deduce how well the method works. Hopefully the authors can clarify some of these issues in the response period.

---

> ### Author Response · Authors · 2017-12-05
> **Re: Interesting idea and good discussion, but unclear method and results**
>
> Hi,
>
> Thank you for your time and evaluation.
>
> I now respond to your queries in the order that they were presented
>
> >1. First, the authors repeatedly state that perturbations are applied to the policy parameters. This seems very strange to me, as typically robust control considers perturbations to the state or control. And reading the actual method, I can't actually figure out how perturbations are applied to the parameters -- as near as I can tell, the perturbations are indeed applied to the controls. So which is it?
>
> Sorry for the confusion. The perturbations are applied to the local controls, p(u|x). Since the policy is trained on all the possible local controllers, the global neural network policies that are learnt through supervised learning are perturbed. We will amend the text to express this more clearly.
>
> > 2.1 There is quite a lot of math in the derivation, and it's unclear which parts relate to the standard guided policy search algorithm, and which parts are new.
>
>
> The math in the derivations (mostly appendices I - II) relate to the trajectory optimization phase of the algorithm. The recursions for the value function and Q functions are now slightly more complicated due to the presence of the adversarial perturbation term. Most of these math is buried in the appendix.
>
> > 2.2 After reading the technical sections several times, my best guess is that the method corresponds to using an adversarial trajectory optimization setup to generate supervision for training a policy. So only the trajectory optimization phase is actually different. Is that true? Or are there other modifications? Some sort of summary of the overall method would have been appreciated, or else a clearer separation of new and old components.
>
> Yes, you are absolutely correct. We reformulate the trajectory optimization phase of the GPS algorithm in a two-player Markov decision process  (with adversarial components to the Q-function expansion in equation 7). We then cast the optimization as an alternating best response supervised learning update of global control and adversary policies to obtain convergence to saddle-point equilibrium. The major difference is the trajectory optimization update but if you look in algorithm II, the C-Step involves a min-max over the augmented cost function ell(x_t, u_t, v_t) and not just the cost function min_{pi \in \Pi} max_{\mu \in M} \ell_t(x_t, u_t) so that we end up with a joint stage cost function \ell_t(x_t, u_t, v_t) = c(x_t, u_t) - \gamma \alpha(v_t), where \alpha is a 2-norm in our implementation.
>
>
> > 3. The evaluation also leaves a lot to be desired. What kind of perturbations are actually being considered? Are they all adversarial perturbations? Do the authors actually test model mismatch or other more natural conditions where robustness would be beneficial? In the end, I was unable to really interpret what the experiments are trying to get across, which makes it hard for me to tell if the method actually works or improves on anything.
>
> Thanks for asking. For our evaluations, we considered additive adversarial perturbation terms introduced by the gamma disturbance parameter on the joint stage cost (eq. 4). The result for testing various adversarial perturbations are in figure 2 for the peg insertion task.
>  	*  as γ --> 1, optimal adversary policy does nothing
>  	*  as γ decreases, adversary actions have larger effect on closed-loop system
>  	*  smallest γ where adversarys policy causes unacceptable performance provides measure
> of robustness of control policy π
> 	*  any existing (deep) RL method can be used to train adversary policy

---

### Official Review · AnonReviewer2 · 2017-11-27
**Promising approach, but oversold and lacking good experimental evidence.**

**Rating:** 3
**Confidence:** 3

**Review:**

The paper presents a method for evaluating the sensitivity and robustness of deep RL policies, and proposes a dynamic game approach for learning robust policies.

The paper oversells the approach in many ways. The authors claim that "experiments confirm that state-of-the-art reinforcement learning algorithms fail in the presence of additive disturbances, making them brittle when used in situations that call for robustness". However, their methods and experiments are only applied to Guided Policy Search (GPS), which seems like a specialized RL algorithm. Conclusions drawn from empirically running GPS on a problem cannot be generalized to all "state-of-the-art RL algorithms".

In Fig 3, the authors conclude that "our algorithm uses lesser number for the GMMs and requires fewer samples to generalize to the real-world". I'm not sure how this can be concluded from Fig 3 [LEFT]. The two line graphs for different values of gamma almost overlay each other, and the cost seems to go up and down, even with number of samples on a log scale. If this shows the variance in the procedure, then the authors should run enough repeats of the experient to smooth out the variance and show the true signal (with error bars if possible). All related conclusions with regards to the dynamic game achieving higher sample efficiency for GMM dynamics fitting need to be backed up with better experimental data (or perhaps clearer presentation, if such data already exists).

Figures 2 and 3 talk about optimal adversarial costs. The precise mathematical definition of this term should be clarified somewhere, since there are several cost functions described in the paper, and it's unclear which terms are actually being plotted here.

The structure of the global policies used in the experiments should be mentioned somewhere.

Note about anonymity: Citation [21] breaks anonymity, since it's referred to in the text as "our abstract". The link to the YouTube video breaks author anonymity. Further, the link to a shared dropbox folder breaks reviewer anonymity, hence I have not watched those videos.

---

### Official Review · AnonReviewer1 · 2017-11-28

**Rating:** 5
**Confidence:** 2

**Review:**

There are two anonymity violations in the paper. The first is in the sentence "The 7-DoF robot result presented shortly previously appeared in our abstract that introduced robust GPS [21]". The second is in the first linked video, which links to a non-anonymized youtube video. The second linked video, a dropbox link, does not have the correct permissions set, and thus cannot be viewed. Also, the citation style does not seem to follow the ICLR style guidelines.

Disregarding the anonymity and style violations, I will review the paper.  I do not have background in H_inf control theory, but I will review the paper to the best of my ability.

This paper proposes a guided policy search method for training deep neural network policies that are robust to worst-case additive disturbances. To my knowledge, the approach presented in the paper is novel, though some relevant references are missing. The experimental results demonstrate the method on two simulated experimental domains, demonstrating robustness to adversarial perturbations. The paper is generally well-written, but has some bugs and typos. The paper is substantially longer than the strongly suggested page limit. There are parts of the paper that should be moved to an appendix to accommodate the page limit.

Regarding the experiments:
My main concerns are with regard to the completeness of the experiments. First, the experimental results report performance in terms of cost/reward, which is extremely difficult to interpret. It would be helpful to also provide success rate for all experiments, where the authors can define success as, e.g. getting the peg in the hole or being within a certain threshold of the goal.
Second, the paper should provide a comparison of policy robustness between the proposed approach and (1) a policy trained with standard GPS, (2) a policy trained with GPS and random perturbations, and ideally, (3) prior approaches to robustness, e.g. Pinto et al., Madelkar et al. [1], or Rajeswaran et al. [2].

Regarding related work and clarity:
There are a few papers that consider the problem of building deep neural network policies that are robust [1,2] that should be discussed and ideally compared to.
Recent deep reinforcement learning work has studied the problem of robustness to adversarial perturbations in the observation space, e.g. [3,4,5,6]. As such, it would be helpful to clarify in the introduction that this paper is considering additive perturbations in the action space.
The paper switches between using rewards and costs. It would be helpful to pick one term and stick with it for the entire paper, rather than switching. Further, it seems like there are errors due to the switching. e.g. on page 3, \ell is defined as the expected reward and in equation 3, it seems like the protaganist policy is trying to minimize \ell, contradicting the earlier definition.
Lastly, section 5.1 is currently rather difficult to follow. It would help to see more top-down direction in the derivation and more details in section 5.1, 5.2, and 5.3 to be moved to an appendix.

Regarding correctness:
There seem to be some issues in the math and/or notation:
The most major issue is in Algorithm 2, which is probably the most important part of the paper to be correct, given that it provides a complete picture of the algorithm. I believe that steps 3 and 4 are incorrect and/or incomplete. Step 4 should be referring to the local policy p rather than the global policy pi (assuming the notation in sections 5.1 and 5.3). Also, the local policy p(v|x) appears nowhere in the algorithm, and probably should appear in step 4. In step 5, how is this regression different from step 7?
In equation 1 and elsewhere in section 4, there is a mix of notation, using u and z inter-changably. (e.g. in equation 1 and the following equation, I believe the u should be switched to z or the z should be switched to u).

Minor feedback:
> "requires fewer samples to generalize to the real world"
None of these experiments are in the real world, so the term "real world" should not be used.
> "algoruithm" -> "algorithm"
> "when unexpected such as when there exists" - bad grammar / typo
> "ertwhile sensy-sensitivity parameter" - typo
- reference 1 is missing authors

In summary, I think the paper would be significantly improved with further experimental comparisons, further discussion of related work, and clarifications/corrections on the notation, equations, and algorithms.  In its current form, I don't think the paper is fit for publication. My rating is between a 4 and a 5.

[1] http://vision.stanford.edu/pdf/mandlekar2017iros.pdf
[2] https://arxiv.org/abs/1610.01283
[2] https://arxiv.org/abs/1701.04143
[3] https://arxiv.org/abs/1702.02284
[4] https://www.ijcai.org/proceedings/2017/0525.pdf
[5] https://arxiv.org/abs/1705.06452

---

### Decision · Program_Chairs · 2018-01-29
**ICLR 2018 Conference Acceptance Decision**

**Decision:**

Reject

**Comment:**

The reviewers are unanimous that the paper is not sufficiently clear and could be improved with better empirical results.